# Prediction of One-Year Mortality Based upon A New Staged Mortality Risk Model in Patients with Aortic Stenosis Undergoing Transcatheter Valve Replacement

**DOI:** 10.3390/jcm8101642

**Published:** 2019-10-08

**Authors:** Verena Veulemans, Amin Polzin, Oliver Maier, Kathrin Klein, Georg Wolff, Katharina Hellhammer, Shazia Afzal, Kerstin Piayda, Christian Jung, Ralf Westenfeld, Alexander Blehm, Artur Lichtenberg, Malte Kelm, Tobias Zeus

**Affiliations:** 1Division of Cardiology, Pulmonology and Vascular Medicine, Heinrich Heine University, Medical Faculty, Moorenstr. 5, 40225 Düsseldorf, Germany; Amin.Polzin@med.uni-duesseldorf.de (A.P.); Oliver.Maier@med.uni-duesseldorf.de (O.M.); Kathrin.Klein@med.uni-duesseldorf.de (K.K.); Georg.Wolff@med.uni-duesseldorf.de (G.W.); Katharina.Hellhammer@med.uni-duesseldorf.de (K.H.); Shazia.Afzal@med.uni-duesseldorf.de (S.A.); Kerstin.Piayda@med.uni-duesseldorf.de (K.P.); Christian.Jung@med.uni-duesseldorf.de (C.J.); Ralf.Westenfeld@med.uni-duesseldorf.de (R.W.); Malte.Kelm@med.uni-duesseldorf.de (M.K.); Zeus@med.uni-duesseldorf.de (T.Z.); 2Division of Cardiovascular Surgery, Heinrich Heine University, Medical Faculty, Moorenstr. 5, 40225 Düsseldorf, Germany; Alexander.Blehm@med.uni-duesseldorf.de (A.B.); Artur.Lichtenberg@med.uni-duesseldorf.de (A.L.); 3CARID (Cardiovascular Research Institute Düsseldorf), Moorenstr. 5, 40225 Düsseldorf, Germany

**Keywords:** outcome, risk scores, TAVR

## Abstract

Background: In-depth knowledge about potential predictors of mortality in transcatheter aortic valve replacement (TAVR) is still warranted. Currently used risk stratification models for TAVR often fail to reach a holistic approach. We, therefore, aimed to create a new staged risk model for 1-year mortality including several new categories including (a) AS-entities (b) cardiopulmonary hemodynamics (c) comorbidities, and (d) different access routes. Methods: 737 transfemoral (TF) TAVR (84.3%) and 137 transapical (TA) TAVR (15.7%) patients were included. Predictors of 1-year mortality were assessed according to the aforementioned categories. Results: Over-all 1-year mortality (*n* = 100, 11.4%) was significantly higher in the TA TAVR group (TF vs. TA TAVR: 10.0% vs. 18.9 %; *p* = 0.0050*). By multivariate cox-regression analysis, a three-staged model was created in patients with fulfilled categories (TF TAVR: *n* = 655, 88,9%; TA TAVR: *n* = 117, 85.4%). Patients in “stage 2” showed 1.7-fold (HR 1.67; CI 1.07–2.60; *p* = 0.024*) and patients in “stage 3” 3.5-fold (HR 3.45; CI 1.97–6.05; *p* < 0.0001*) enhanced risk to die within 1 year. Mortality increased with every stage and reached the highest rates of 42.5% in “stage 3” (*p*_logrank_ < 0.0001*), even when old- and new-generation devices (*p*_logrank_ = n.s) were sub-specified. Conclusions: This new staged mortality risk model had incremental value for prediction of 1-year mortality after TAVR independently from the TAVR-era.

## 1. Introduction

Transcatheter aortic valve replacement (TAVR) has proven to be an effective technique in patients with symptomatic aortic stenosis (AS). Despite TAVR-related improvement, mortality and limited clinical benefits are still observed, especially in patients undergoing transapical TAVR (TA TAVR). Transapical patients especially show higher mortality as compared to transfemoral patients [1,2]. Furthermore, different AS-entities by the meaning of high-gradient (HG), paradoxical, and true low-gradient ((*p*)LG) aortic stenosis are well-known to be associated with different outcomes [3]. The new updated recommendations of pulmonary hypertension (PHT) associated with left heart disease gives further overview of sub-classifications of PHT leading to enhanced mortality in patients with AS [4]: In patients undergoing TAVR, pre-interventional PH is associated with a poor prognosis [5], and especially precapillary and combined entities of PHT seem to be associated with a significantly higher 1-year mortality [6,7]. However, currently used risk stratification models for TAVR are by the majority based on the presence of comorbidities [8,9], and fail to reach a holistic approach including the functional status, hemodynamic patterns, and comorbidities.

Therefore, we sought to (i) analyze the influence of AS-entities, cardiopulmonary hemodynamics, comorbidities, and different access routes on 1-year mortality (ii) create a new staged risk model based on these factors and (iii) assess the impact of the resulting staged risk model on survival. 

## 2. Experimental Section

From 2009 to 2018, a total of 874 patients with either transfemoral (TF, *n* = 737, 84.3%) or transapical (TA, *n* = 137, 15.7%) TAVR and complete hemodynamic status were retrospectively enrolled and were included in this analysis. All procedures were performed according to current guidelines and under local or general anesthesia. TF TAVR was performed with different generations of either the self-expandable CoreValve System (Medtronic Inc., Minneapolis, MN, USA) or the balloon-expandable SAPIEN System (Edwards Lifesciences, Irvine, CA, USA). TA TAVR was predominantly performed by using the SAPIEN System (Edwards Lifesciences, Irvine, CA) or in very few cases the Engager System (Medtronic Inc., Minneapolis, MN, USA). Old-generation devices included the SAPIEN XT^TM^, CoreValve^TM^, Engager^TM^ and Jena Valve^TM^. New-generation devices covered the SAPIEN 3^TM^, CoreValve Evolut R^TM^, and CoreValve Evolut Pro^TM^. All patients provided written informed consent for TAVR and the use of clinical, procedural and follow-up data for research. The study procedures were in accordance with the Declaration of Helsinki and the institutional Ethics Committee of the Heinrich-Heine University approved the study protocol (4080). The study is registered at clinical trials (NCT01805739). The study endpoint was defined as 1-year all-cause mortality. 

High-gradient AS (HG-AS) was defined as normal left-ventricular function (LVF) >50% with high gradients (mean gradient > 40 mmHg). Paradoxical LG-AS (pLG-AS) was defined as preserved LVF > 50% combined with a mean gradient <40mmHg according to current guideline definitions. Low-gradient aortic stenosis (LG-AS) was defined as reduced LVF <50% combined with a mean gradient <40 mmHg. 

Patients underwent coronary angiography and hemodynamic assessment prior to TAVR according to current recommendations. Systolic, diastolic, mean pulmonary artery pressure (mPAP) and pulmonary capillary wedge pressure (PCWP) were measured. Cardiac output (CO) was assessed by the indirect Fick method. The transpulmonary gradient (TPG) was calculated as mPAP–PAWP. LVEDP was recorded after crossing the aortic valve. Pulmonary hypertension (PHT) was defined as mPAP ≥ 25 mmHg and was classified as pre-capillary (precPHT; PCWP ≤ 15 mmHg), isolated post-capillary (IpcPHT; PCWP > 15 mmHg, PVR ≤ 3 Wood units (WU)), or combined entity (cpcPHT; PCWP > 15 mmHg, PVR > 3WU) according to existing guidelines [10].

The collected data included patient characteristics, imaging findings, periprocedural in-hospital data, laboratory results and follow-up data. Follow-up data were obtained on the basis of out-patient follow-up or follow-ups according to longitudinal re-hospitalizations. Continuous data were described by mean and standard deviation, median or upper and lower 95% confidence interval (interquartile ranges) and categorical variables by frequencies and percentages. Continuous variables were compared using a Student’s *t*-test or Kolmogorov-Smirnov test depending on variable distribution in a heterogenous sample size. Categorical variables were compared using Fishers’ exact test. Survival was compared using Kaplan–Meier plots and log-rank tests. Cox regression was performed to assess independent predictors of mortality. Covariates associated with mortality in the univariate analysis (*p* < 0.1) were entered into the multivariate model. Model discrimination accuracy was evaluated using ROC analysis and the *c*-index (area-under-the-curve) as a cumulative measure. Predictors with a HR of 1.00 to 1.74 received 1 point, those with a HR between 1.75 to 2.49 received 2 points and those above a HR of 2.50 were assigned 3 points. 

The data analysis was performed using the statistical software SPSS (version 23.0, SPSS Inc., Chicago, IL, USA) and GraphPad Prism (version 6.0, Graphpad Software, San Diego, CA, USA). All statistical tests were 2-tailed, and a value of *p* < 0.05 was considered statistically significant. 

## 3. Results

### 3.1. Baseline Characteristics—Clinical and Functional Data

As expected, baseline patients’ characteristics did differ related to the particular risk profile and selection bias of transfemoral versus transapical assigned approaches: Transapical patients were younger (TF vs. TA AVR: Age 80.9 ± 6.0 vs. 78.5 ± 6.3; *p* = 0.0013*) and predominantly male (TF vs. TA TAVR: Female 56.2% vs. 40.2%; *p* = 0.0005*). In summary, general atherosclerosis in the meaning of concomitant coronary artery disease (CAD), peripheral artery disease (PAD), cerebrovascular disease (CVD) and porcelain aorta (PAo) were more frequent in patients undergoing TA TAVR. Functional parameters did not differ except of more frequent 2nd degree aortic regurgitation (TF vs. TA TAVR: AR 15.3% vs. 23.4%; *p* = 0.0243*), higher logistic EuroSCORE I (logES-I) and STS-PROM according to the pre-described risk profile (TF vs. TA TAVR: logES-I 26.4 ± 16.8 vs. 30.8 ± 17.6; *p* = 0.0011*; STS-PROM 6.7 ± 6.5 vs. 7.7 ± 7.0; *p* = 0.0217*). A full overview of baseline clinical and functional characteristics is displayed in the Table A1.

### 3.2. Outcome Analysis In Different TAVR Approaches

After one year, there were 100 deaths (11.4%) in the overall cohort, resulting in a mortality distribution of 10.0% in the TF TAVR and 18.9% in the TA TAVR cohort. Periprocedural death within the first 30 days was recorded in 31 cases (TF vs. TA TAVR: 2.8% vs. 8.0 %; *p* = 0.0048*). Cardiovascular death was documented in 40 cases (TF vs. TA TAVR: 3.8% vs. 8.8 %; *p* = 0.0227*). Further causes of over-all death after 1 year were cancer (*n* = 2; 0.2%), cerebrovascular disease (*n* = 5, 0.6%), infection/sepsis (*n* = 19; 2.2%), polytrauma (*n* = 1; 0.1%), comorbidities leading to all-cause death (*n* = 4; 0.5%) and unknown reasons (*n* = 29; 3.3%). A full overview of baseline clinical and functional characteristics is displayed in Table A2.

### 3.3. Predictors of One-Year Mortality

Predictors of 1-year mortality were assessed according to the following categories: AS-entities, cardiopulmonary hemodynamics, comorbidities, and different access routes and are shown in Figure 1. Predictors of death according to the univariate and multivariate analysis are offered in Table 1.

#### 3.3.1. AS-Entities

According to underlying filling pressure pathology, all patients were stratified into subgroups of AS-entity (HG-AS, pLG-AS and LG-AS). By multivariate cox regression analysis, only LG-AS (HR 2.06; CI 1.28–3.32; *p* = 0.003*) was associated with increased 1-year mortality.

#### 3.3.2. Cardiopulmonary Hemodynamics

From all hemodynamic patterns tested by univariate analysis, including PHT sub-classifications and moderate-to-severe valvular diseases, only a supra-median PAPm ≥28 mmHg (HR 1.53; CI 1.01–2.31; *p* = 0.044*) and moderate-to-severe tricuspid regurgitation (TR) (HR 1.98; CI 1.28–3.06; *p* = 0.002*) were associated with mortality. Neither over-all PHT, precPHT, ipcPHT or cpcPHT, LVEDP, cardiac index or PCWP alone were independent predictors for 1-year mortality.

#### 3.3.3. Other Cofactors and Comorbidities

Oral anticoagulation (HR 1.53; CI 1.03–2.28; *p* = 0.037*), renal replacement therapy (HR 2.93; CI 1.68–5.12; *p* < 0.0001*) and diabetes mellitus (HR 1.50; CI 1.01-2.25; *p* = 0.047*) were independent risk factors for 1-year mortality. 

#### 3.3.4. Access Routes

TA TAVR (HR 1.79; CI 1.14–2.82; *p* = 0.001*) was associated with a nearly 2-fold enhanced risk for 1-year death.

### 3.4. Staged Risk Model for One-Year Mortality in TAVR

The characteristics listed in Figure 1A were identified as the best values for prediction of 1-year mortality according to the multivariate cox regression model. In 655 TF TAVR (88.9%) and 117 TA TAVR (85.4%) patients with complete data for risk stratification, a staged risk model including the categories (a) AS-entities (b) cardiopulmonary hemodynamics (c) comorbidities, and (d) different access routes was created by attributing 1–3 points as described above. The risk model with summation of all points can be only calculated when all data of the several categories are available, so the patients had to be reduced for the staged classification analysis due to the completeness of all incorporated determinants. This resulted in a model with a minimum of 1 and a maximum of 10 points. In c-statistics the risk score showed a moderate discrimination for 1-year mortality with an area under the curve of 0.68 (95% CI, 0.62–0.75; *p* < 0.0001*; Figure 1B). Consequently, summation of the aforementioned points within the mortality risk model resulted in a statistically significant difference between survivors and deceased patients (3.1 ± 1.8 vs 4.5 ± 2.2; *p* < 0.0001*, Figure 1C).

Patients were categorized into three different stages according to the number of points (Stage 1: ≤3 points; Stage 2: Points 4–6; stage 3: ≥7 points; Figure 1) depending on the presence or absence of the aforementioned characteristics fulfilling the several categories. Kaplan-Meier curves were plotted to clarify impact of the several stages on event-free survival and 1-year mortality. Mortality increased with every stage and reached the highest overall mortality-rates of 42.5%, classified with more than cumulative 7 points (*p*_logrank_ < 0.0001*, Figure 2A). Looking at old-generation devices (*p*_logrank_ = 0.0050*, Figure 2B) and new-generation devices (*p*_logrank_ < 0.0001*, Figure 2C), no significant difference in mortality distribution according to the three-staged model was detected.

The staged model of mortality had incremental value over other covariates for prediction of 1-year mortality after TAVR: “Stage 1” was associated with reduced mortality (HR 0.38; CI 0.24–0.59; *p* < 0.0001*), whereas patients in “stage 2” showed 1.7 fold (HR 1.67; CI 1.07–2.60; *p* = 0.024*) and patients in “stage 3” 3.5 fold (HR 3.45; CI 1.97–6.05; *p* < 0.0001*) enhanced risk, each with equal distributions for TF and TA TAVR. For further details please see Table 2.

## 4. Discussion

The present study evaluating the influence of AS-entities, cardiopulmonary hemodynamics, comorbidities, and different access routes on 1-year mortality revealed several findings:

(1) LG-AS, a supra-median PAPm ≥28 mmHg, moderate-to-severe tricuspid regurgitation, oral anticoagulation, renal replacement therapy, diabetes mellitus, and transapical approach were independent risk factors for 1-year mortality. 

(2) A three-staged risk model had incremental value over other covariates for prediction of 1-year mortality.

### 4.1. Detailed Outcome Analysis 

Our study demonstrated that transapical patients showed higher mortality than transfemoral patients, being in line with the current knowledge [1,2]. In our study, 1-year mortality was multifactorial, supposing that higher all-cause morbidity in the TA TAVR cohort leads to the well-known mortality distribution. Currently used risk stratification models for TAVR are by the majority based on the presence of comorbidities [8,9] and fail to reach a holistic approach, including the functional status, hemodynamic patterns, and comorbidities. Genereux et al. [11] was the first group including the extent of cardiac damage into a new risk model, demonstrating that consideration of AS-related cardiac damage in future recommendations for risk stratification might be useful. AS-entities such as HG-, pLG-, and LG-AS with different filling pathologies have been described to be associated with different outcomes [3]. To date, still few data exist on impact and (poor) prognosis of LG-AS patients undergoing TAVR. Dobutamine stress echocardiography to determine contractile reserve is questionable to predict outcome in this unique patient cohort [12]. In our study—as expected—only LG-AS was associated with enhanced 1-year mortality, whereas cardiac index alone had no impact, supposing that pronounced cardiac and structural damage as global myocardial impairment might be an essential predictor to integrate into future risk prediction models.

In patients undergoing TAVR, pre-interventional PHT is well-known to be associated with a poor prognosis [5], and especially precapillary and combined entities of PHT seem to be associated with a significantly higher 1-year mortality [6,7]. However, in our study, neither PHT in general, precapillary, isolated or combined entities of PHT were independent predictors for 1-year mortality, whereas a supra-median PAPm ≥28 mmHg was strongly and independently associated with enhanced mortality, suggesting that only advanced pulmonary hypertension independent from the underlying mechanism may have a real impact on mortality. 

Furthermore, moderate-to-severe tricuspid regurgitation as a marker for impaired right heart function was a strong predictor of 1-year death. The impact of tricuspid regurgitation on outcome in patients undergoing TAVR is poorly defined until now and therefore, not regularly taken into consideration for risk stratification [13]. However, advanced right ventricular dysfunction might be a late and adverse sign of cardiac functionality secondary to left ventricular dysfunction, left ventricular hypertrophy, pulmonary disease or primary independent ventricular mechanisms [11], potentially closing the circle to different outcomes in patients with LG-AS.

Interestingly, oral anticoagulation, but not atrial fibrillation was an independent risk factor for 1-year mortality. AF had some influence by univariate analysis but failed to be an independent predictor for 1-year mortality by multivariate analysis. Several studies showed that pre-existing and new-onset AF had been associated with higher rates of mortality at one year in TAVR patients [14,15].

In our study, patients under OAC were predominantly treated with Marcumar, so we think that the enhanced influence on 1-year mortality is driven by a comorbidity summation effect: Patients on Marcumar were more frequently under RRT or had surgical valve replacement, and had more history of generalized atherosclerosis. Even if the multivariate analysis discriminates on this factor, there is a big chance that OAC therapy also maps other independent risk factors like RRT and AF itself.

### 4.2. Staged Risk Model for 0ne-Year Mortality in TAVR

Notably, we demonstrated that a staged risk model including the categories (a) AS-entities (b) cardiopulmonary hemodynamics (c) comorbidities, and (d) different access routes had incremental value over other covariates for prediction of 1-year. Multiple hemodynamic and comorbidity-based covariables have been adjusted by multivariate regression analysis. However, after adjustment, only LG-AS, a supra-median PAPm ≥28 mmHg, moderate-to-severe tricuspid regurgitation, transapical approach, oral anticoagulation, renal replacement therapy, and diabetes mellitus remain independent risk factors for mortality. Depending on the presence or absence of the characteristics mentioned above fulfilling the several categories, three independent stages according to the obtained number of points (Stage 1: ≤3 points; Stage 2: points 4–6; stage 3: ≥7 points; Figure 1) were created. “Stage 1” was associated with low 1-year mortality, whereas patients in “stage 2” showed 1.7-fold and patients in “stage 3” 3.5-fold enhanced risk, each with equal distributions for TF and TA TAVR, suggesting again that not transapical approach alone but treatment modality in combination with preserved or impaired functional status and comorbidities might be the real driver for early mortality: 1-year mortality increased with every stage and reached the highest rates of 42.5% in patients classified with more than cumulative 7 points. Interestingly, TAVR treatment era comparing old- and new-generation devices had no impact on mortality related to our three-staged risk model, emphasizing a universal validity independent from device-generation, knowledge, and presumably classical risk scoring. It is important to note that the majority of patients died of unknown or multifactorial reasons including septic conditions, respiratory diseases, and cerebrovascular events, suggesting that advanced cardiac impairment in addition to adverse comorbidities define a vulnerable patient collective at high-risk for the impaired outcome.

To our knowledge, this is the first study addressing consideration of AS-entity with varying flow-patterns, cardiopulmonary hemodynamic status, access routes, and comorbidities as meaningful variables that could be easily incorporated into a simple risk-prediction model. 

### 4.3. Limitations

Outcomes were only available up to one year only, and further larger studies are needed to prospectively validate this staged mortality risk model in lower risk populations with possibly changing comorbidity impact: With only 100 deceased patients in this study cohort and the underlying three-staged classification, no validation cohort can be offered in terms of power and validity. As a result, a major limitation of this study is based on the fact that the same cohort was established for both -the prediction and test model. The increased mortality in TA TAVR is probably due to selection bias, although the multivariate analysis balanced this fact, and TA TAVR was allocated with two points within the staged risk model leading to a weighted mortality distribution within this model. Furthermore, echocardiographic right heart function parameters are limited. For a more detailed consideration of right heart-derived mortality aspects, further parameters like TAPSE, TEI-index, and right heart volumes should be included in the future. 

## 5. Conclusions

This new staged mortality risk model including (a) AS-entities (b) cardiopulmonary hemodynamics (c) comorbidities, and (d) different access routes had incremental value for prediction of 1-year mortality after TAVR independently from the TAVR-era.

## Figures and Tables

**Figure 1 jcm-08-01642-f001:**
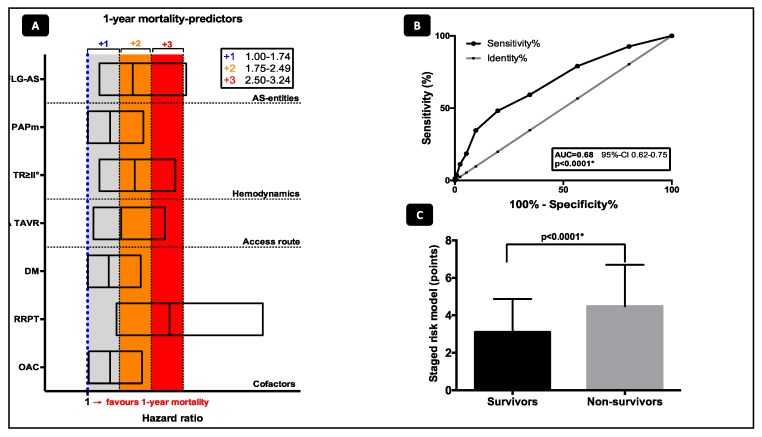
(**A**) Awarding of points based on the aforementioned HR calculation for predictors of 1-year mortality according to the categories (a) AS-entities (b) cardiopulmonary hemodynamics (c) comorbidities, and (d) different access route in a multivariate cox-regression adjustment model: Predictors with a HR of 1.00 to 1.74 received 1 point, those with a HR between 1.75 to 2.49 received 2 points and those above a HR of 2.50 were assigned 3 points. (**B**) Area under the curve (AUC) as a receiver operating characteristic (ROC) by c-statistic for discriminating performance of the staged risk model. (**C**) Cumulative calculation points comparing survivors and non-survivors.

**Figure 2 jcm-08-01642-f002:**
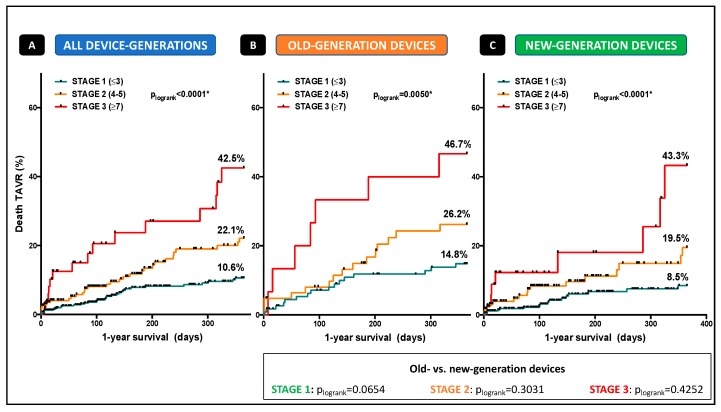
Kaplan-Meier survival curves for 1-year mortality according to the staged risk stratification and device-generation. (**A**) Kaplan-Meier survival curves according to the staged risk stratification in the overall cohort (**B**) using the old-generation, and (**C**) new-generation devices.

**Table 1 jcm-08-01642-t001:** Univariate and multivariate Cox regression analysis with mortality as the dependent variable.

1-Year Mortality	Univariate Analysis		Multivariate Analysis	
(A) AS-Entities	Hazard Ratio (95-CI)	*p*-Value	Hazard Ratio (95-CI)	*p*-Value
HG-AS	0.90 (0.56–1.46)	0.671	-	
pLG-AS	0.66 (0.42–1.02)	0.061	-	
LG-AS	2.06 (1.28–3.32)	0.003 *	2.06 (1.28–3.32)	0.003 *
**(B) Hemodynamics**				
PCWP ≥ 20 mmHg	1.67 (1.12–2.47)	0.011 *	-	
PAPm ≥ 28 mmHg	1.65 (1.09–2.48)	0.017 *	1.53 (1.01–2.31)	0.044 *
TR ≥ II°	2.10 (1.36–3.24)	0.001 *	1.98 (1.28–3.06)	0.002 *
**(C) Other cofactors**				
TA approach	1.92 (1.23–3.01)	0.004 *	1.79 (1.14–2.82)	0.012 *
BMI	1.00 (1.00–1.00)	0.050	-	
PAD	1.72 (1.15–2.55)	0.008 *	-	
Porcelain Aorta	1.78 (1.09–2.91)	0.020 *	-	
OAC	1.54 (1.03–2.28)	0.034 *	1.53 (1.03–2.28)	0.037 *
RRPT	3.20 (1.85–5.54)	<0.0001 *	2.93 (1.68–5.12)	<0.0001 *
DM	1.63 (1.10–2.42)	0.015 *	1.50 (1.01–2.25)	0.047 *
AF	1.46 (0.98–2.19)	0.067	-	

* *p* < 0.05 was considered statistically significant. Univariate and multivariate analysis of several baseline conditions and their impact on mortality (estimated as Hazard ratio with 95%-CI). HG-AS = normal-flow high-gradient AS; (p)LG-AS = (paradoxical) low-flow low-gradient AS; PCWP = pulmonary wedge pressure; PAPm = pulmonary artery pressure (mean); TR = tricuspid regurgitation. BMI = body mass index; PAD = peripheral artery disease; DM = diabetes mellitus; OAC = oral anticoagulation; RRPT = renal replacement therapy; AF = atrial fibrillation.

**Table 2 jcm-08-01642-t002:** Details of staged mortality risk models.

1-Year Mortality	Approach			Univariate Analysis	
(A) STAGES	TF TAVR	TA TAVR	*p*-Value	Hazard Ratio (95-CI)	*p*-Value
STAGE 1 (≤3 points) (33/485; 10.6%)	29/441	4/44	0.5262	0.38 (0.24–0.59)	<0.0001 *
STAGE 2 (4–6 points) (33/235; 22.1%)	22/177	11/58	0.2752	1.67 (1.07–2.60)	0.024 *
STAGE 3 (≥7 points) (15/52; 42.5%)	9/37	6/15	0.3181	3.45 (1.97–6.05)	<0.0001 *

* *p* < 0.05 was considered statistically significant. Values are % (*n*). p-values of mortality within the stages are estimated by Kaplan–Meier method, and do not account for competing risks. Univariate analysis of the several stages and their impact on mortality are estimated as Hazard ratio with 95%-CI.

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
