# Peer review of "Prediction of One-Year Mortality Based upon A New Staged Mortality Risk Model in Patients with Aortic Stenosis Undergoing Transcatheter Valve Replacement"

_jcm, 2019, doi:10.3390/jcm8101642_

Round 1

Reviewer 1 Report

The authors performed a retrospective study to investigate the association between a novel risk stratification model and 1-year mortality after TAVI. The analysis was conducted in 874 TAVI patients (737 TF-TAVI and 137 TA-TAVI patients). The study revealed a significantly higher 1-year mortality rate in the TA TAVI group. By multivariate cox-regression a three-staged risk model was created which showed an incremental association of every stage with 1-year mortality after TAVI. While this interesting study clearly addresses a very important issue in the context of TAVI, there are some areas of improvement:   - Study population: As pointed out by the authors, TA TAVR patients in general have a higher mortality rate than TF patients and differ from TF patients in numerous aspects. This, however, raises the question why the study population included TA TAVR patients in the first place, as this might lead to substantial confounding when deriving mortality-associated risk factors. The number of TF patients (n=737) seems to be high enough to develop a risk model in TF patients only. Have the authors performed such an analysis?   - AS entities: The proposed risk model includes different AS entities. Patients were therefore stratified into 3 different subgroups of AS (NF-HG, paradoxical LF-LG and classical LF-LG). However, the authors should also address, include and discuss the issue of normal-flow low-gradient AS (NF-LG), which is present in an important subgroup of TAVI patients.   - Table S1. Patient Clinical and Functional Characteristics: Clincial data: Please include parameters of renal function (eGFR and/or creatinine) and echocardiography data regarding left ventricular systolic function (i.e., LVEF). Also right ventricular function should be reported (e.g., TAPSE). Functional data: Are there other cardiovascular biomarkers available in addition to NT-proBNP (e.g., (hs)CRP, high-sensitivity Troponins)? Please include, if possible.   - Procedural outcome: In addition to 30-day mortality, other major procedure-related outcomes should be presented (VARC2-criteria).   - Statistics: Page 1, line 20/21: […] 737 transfemoral (TF) TAVR (41.2%) and 137 transapical (TA) TAVR (7.7%) with complete hemodynamics […]. - 41.2%/7.7% seems to be incorrect. Please correct/explain. Page 2, line 53/54: […] a total of 874 patients with either transfemoral (TF, n=737, 41.2%) or transapical (TA, n=137, 7.7%) TAVR […]. - Please see above. Page 5, line 152/153: […] In 655 TF TAVR and 117 TA TAVR patients with complete data […]. - Please explain the missing 82 TF (737-655) patients.   - Cofactors and comorbidities: Does “renal replacement therapy” (RRT) refer to chronic RRT or new RRT after TAVI? Please specify. Do the authors have an explanation why oral anticoagulation, but not AF was significantly associated with 1-year mortality?   - Follow-up: Please specify how the follow-up data were obtained. Are follow-up data beyond 1 year available? If so, Kaplan-Meier curves >1 year would be an interesting addition.   - Limitations: The authors should include a brief limitations section.

Author Response

Dear Editors,

dear Reviewers,

we thank you very much for your in-depth review of our manuscript “Prediction of 1-year mortality based upon a new staged mortality risk model in patients with aortic stenosis undergoing transcatheter valve replacement” and for giving us the opportunity to submit a revised version of our manuscript.

We have thoroughly revised the manuscript according to the reviewers’ comments and included our detailed replies below for your convenience.

Point-by-point review:

Reviewer #1:

Reviewer:- Study population: As pointed out by the authors, TA TAVR patients in general have a higher mortality rate than TF patients and differ from TF patients in numerous aspects. This, however, raises the question why the study population included TA TAVR patients in the first place, as this might lead to substantial confounding when deriving mortality-associated risk factors. The number of TF patients (n=737) seems to be high enough to develop a risk model in TF patients only. Have the authors performed such an analysis?  

Reply: We would like to thank you for your valuable and helpful comments. You are completely right with your statement of enhanced mortality in TA TAVR patients. Our aim was exactly to build a risk model for both access routes, often under-represented in current risk models, taken several risk factors and hemodynamic circumstances into account. All covariates where tested by uni- and multivariate analysis to balance the confounding between both cohorts. However, we did not perform an analysis on TF alone, because this was not the design and intention of this study.

Reviewer:- AS entities: The proposed risk model includes different AS entities. Patients were therefore stratified into 3 different subgroups of AS (NF-HG, paradoxical LF-LG and classical LF-LG). However, the authors should also address, include and discuss the issue of normal-flow low-gradient AS (NF-LG), which is present in an important subgroup of TAVI patients.

Reply: Thank you very much for your advise.We did only consider and separate HG-, pLG and LG-AS according to the mean gradient and LVF (see also definition page 2, lines 67-71) and did not take flow behaviour into account. We corrected the abbreviations to avoid misunderstandings in the whole document.

Reviewer:- Table S1. Patient Clinical and Functional Characteristics: Clincial data: Please include parameters of renal function (eGFR and/or creatinine) and echocardiography data regarding left ventricular systolic function (i.e., LVEF). Also right ventricular function should be reported (e.g., TAPSE). Functional data: Are there other cardiovascular biomarkers available in addition to NT-proBNP (e.g., (hs)CRP, high-sensitivity Troponins)? Please include, if possible.  

Reply: Thank you for this remark. We added information about GFR, creatinine, Trop T and LVF as staged model and also calculated as LVEF, as well as FeV1 into the S1 table.

Concerning right ventricular function, the retrospective data are very limited to give a valid view and calculation. We argued on this fact in the limitation section. However, pronounced TR is an easy to handle parameter and important prognostic impact factor, what is also mentioned in current TAVR-related outcome studies.

Reviewer:- Procedural outcome: In addition to 30-day mortality, other major procedure-related outcomes should be presented (VARC2-criteria).  

Reply: Thank you again, but we only added 30-day mortality for discrimination towards 1-year mortality. 30-day outcome or secondary endpoints according to VARC-2 was generally not the focus in this study and is reflected by multiple others random samples. However, if you feel that 30-day mortality might be crucial in this context, we will offer the missing data.

Reviewer:- Statistics: Page 1, line 20/21: […] 737 transfemoral (TF) TAVR (41.2%) and 137 transapical (TA) TAVR (7.7%) with complete hemodynamics […]. - 41.2%/7.7% seems to be incorrect. Please correct/explain. Page 2, line 53/54: […] a total of 874 patients with either transfemoral (TF, n=737, 41.2%) or transapical (TA, n=137, 7.7%) TAVR […]. - Please see above. Page 5, line 152/153: […] In 655 TF TAVR and 117 TA TAVR patients with complete data […]. - Please explain the missing 82 TF (737-655) patients.  

Reply: Page 1, line 20/21: […] 737 transfemoral (TF) TAVR (41.2%) and 137 transapical (TA) TAVR (7.7%) with complete hemodynamics […]. Page 2, line 53/54: […] a total of 874 patients with either transfemoral (TF, n=737, 41.2%) or transapical (TA, n=137, 7.7%) TAVR […]. Thank you very much, we recalculated and corrected the frequencies in the whole document.

Page 5, line 152/153: […] In 655 TF TAVR and 117 TA TAVR patients with complete data […]. - Please explain the missing 82 TF (737-655) patients. The risk model with summation of all points can be only calculated when all data of the several categories in all patients are available (e.g missing right heart hemodynamic). This was the case in 655/737 TF and 117/137 TA TAVR patients, leading to a portion >85%. Due to the point-allocation, the factor “0” would have led to an undercalculation of risk, so we decided to give a representative image of valid and entire data. We give a short explanation and frequencies on page 5, lines 157 f.

Reviewer:- Cofactors and comorbidities: Does “renal replacement therapy” (RRT) refer to chronic RRT or new RRT after TAVI? Please specify. Do the authors have an explanation why oral anticoagulation, but not AF was significantly associated with 1-year mortality?

Reply: Thank you again for your remark and important questions: RRT refers to chronic RRT according to the baseline criteria. New renal replacement therapy after TAVR was mostly transient and therefore not considered for influence on 1-year mortality. In general, only baseline criteria pre TAVR was included in this analysis.

AF had some influence by univariate analysis but failed to be an independent predictor for 1-year mortality by multivariate analysis. Patients under OAC were predominantly treated with Marcumar, so we think that the enhanced influence on 1-year mortality is driven by a comorbidity summation effect: patients on Marcumar were frequently under RRT or had surgical valve replacement, and had more history of generalized atherosclerosis (so reasons why they didn’t received NOACs). We added some thoughts into the discussion.

Reviewer:  - Follow-up: Please specify how the follow-up data were obtained. Are follow-up data beyond 1 year available? If so, Kaplan-Meier curves >1 year would be an interesting addition.  

Reply:   Follow-up data were obtained by means of out-patient follow-up or follow-ups according to re-hospitalization. Data >1 year were not longitudinal available due to lost in FU and protection of privacy data with impeded data-update. When patients had died, in the most cases the reason of death was unknown. However, we don’t think that the quality of data after 1-year is good enough to give a real hint on long-term influence.

Reviewer:- Limitations: The authors should include a brief limitations section.

Reply:   We appreciate your statement and added a limitation section with several aspects, that you have just considered before.

Reviewer 2 Report

This is a retrospective analysis of 874 TAVI interventions for determining 1 year all-cause mortality. In addition, groups undergoing either transfemoral and transapical TAVI were compared. In the uni-and multivariate analyses, “AS-entities” (meaning classical, pLFLG or LFLG AS), pulmonary hypertension, comorbidities, and access routes were addressed. The lack of a reliable model for predicting outcome after TAVI is an important shortcoming in patient selection for treatment of AS. Although performed in a large cohort, the analyses performed have several limitations.

How were missing data treated? For example, was pulmonary artery pressures available in all 874 patients? The increased mortality in the transapical group is probably due to selection bias. The authors use the same cohort to establish the factors to be included into their prediction model, and to test the model. This is a major limitation, which precludes any conclusions as to the general applicability of the model. There are several factors of importance for the outcome post TAVI that are not addressed. For example, coronary artery disease, previous cardiac surgery, lung function, symptoms…

Author Response

Dear Editors,

dear Reviewers,

we thank you very much for your in-depth review of our manuscript “Prediction of 1-year mortality based upon a new staged mortality risk model in patients with aortic stenosis undergoing transcatheter valve replacement” and for giving us the opportunity to submit a revised version of our manuscript.

We have thoroughly revised the manuscript according to the reviewers’ comments and included our detailed replies below for your convenience.

Point-by-point review:

Reviewer:How were missing data treated? For example, was pulmonary artery pressures available in all 874 patients?
Reply:Several parameters were not available in the entire cohort including pulmonary artery pressures. That is the reason why the cohort had to be reduced for creating the staged mortality risk model. The risk model with summation of all points can be only calculated when all data of the several categories are available, so the patients had to be reduced for the staged classification analysis due to the completeness of all incorporated determinants.

Reviewer:The increased mortality in the transapical group is probably due to selection bias. The authors use the same cohort to establish the factors to be included into their prediction model, and to test the model. This is a major limitation, which precludes any conclusions as to the general applicability of the model.

Reply:We thank you for your valuable comment and discussed this topic in the limitation section: “(…) With only 100 deceased patients in this study cohort and the underlying three-staged classification, no validation cohort can be offered in terms of power and validity. As a result, a major limitation of this study is based on the fact that the same cohort was established for both -the prediction and test model. The increased mortality in TA TAVR is probably due to selection bias, although the multivariate analysis balanced this fact, and TA TAVR was allocated with 2 points within the staged risk model leading to a weighted mortality distribution within this model.”  However, as calculated by multivariate analysis, TA was identified as independent risk factor and weighted according to the HR 1.79 (1.14-2.82) in this model. This also mentioned by balanced mortality rates in Table 2, where patients with TA and TF TAVR had equalized mortality using the staged risk model 1-3.

Reviewer:There are several factors of importance for the outcome post TAVI that are not addressed. For example, coronary artery disease, previous cardiac surgery, lung function, symptoms…

Reply:We agree with your statement that several factors are known to have an impact on the outcome in TAVR patients, including the factors you have mentioned above. All of these parameters are listed in the supplementary file S1, and coronary artery disease was significantly enhanced in patients undergoing TA TAVR (also pronounced in the baseline results) but was no independent risk factor after further analyses. Previous cardiac surgery was also mentioned by CABG and previous SVR with the same effect undergoing uni- and multivariate analysis. We added FeV1 (Table S1) additive to COPD, both risk factors without differences and without impact on mortality by uni-and multivariate analysis. We already documented pronounced NYHA and CCS stage (III-IV) as symptom classification, but without influence on mortality in this cohort (Table S1).

Round 2

Reviewer 1 Report

My concerns have been adequately discussed.

Reviewer 2 Report

The authors performed a minor revision. The limitations pointed out in the initial review apply also to the revised manuscript.